# Healing Pattern Analysis for Dental Implants Using the Mechano-Regulatory Tissue Differentiation Model

**DOI:** 10.3390/ijms21239205

**Published:** 2020-12-02

**Authors:** Ming-Jun Li, Pei-Ching Kung, Yuan-Wei Chang, Nien-Ti Tsou

**Affiliations:** Department of Materials Science and Engineering, National Chiao Tung University, Hsin-chu 30010, Taiwan; a0916986941@gmail.com (M.-J.L.); peggygong1014@gmail.com (P.-C.K.); aone.chang@gmail.com (Y.-W.C.)

**Keywords:** dental implants, mechano-regulatory, tissue differentiation, healing chamber

## Abstract

(1) Background: Our aim is to reveal the influence of the geometry designs on biophysical stimuli and healing patterns. The design guidelines for dental implants can then be provided. (2) Methods: A two-dimensional axisymmetric finite element model was developed based on mechano-regulatory algorithm. The history of tissue differentiation around eight selected implants can be predicted. The performance of the implants was evaluated by bone area (BA), bone-implant contact (BIC); (3) Results: The predicted healing patterns have very good agreement with the experimental observation. Many features observed in literature, such as soft tissues covering on the bone-implant interface; crestal bone loss; the location of bone resorption bumps, were reproduced by the model and explained by analyzing the solid and fluid biophysical stimuli and (4) Conclusions: The results suggested the suitable depth, the steeper slope of the upper flanks, and flat roots of healing chambers can improve the bone ingrowth and osseointegration. The mechanism related to solid and fluid biophysical stimuli were revealed. In addition, the model developed here is efficient, accurate and ready to extend to any geometry of dental implants. It has potential to be used as a clinical application for instant prediction/evaluation of the performance of dental implants.

## 1. Introduction

Dental implants have been widely used to restore tooth function and esthetics. The success of surgeries is affected by the design of implant shapes, loading conditions, and bone/tissue differentiation around the implant [1]. The healing chamber is defined as the space between the adjacent threads [2]. Proper healing chamber design improves the osseointegration around the implant, and reduces the healing period. Beutel et al. [3] discovered that implants with a trapezoidal healing chamber had the best osseointegration among the cases they considered. Marin et al. [2] designed healing chambers with simple geometrical design, and compared the resulting bone ingrowth through histomorphologic evaluation.

The appropriate design of the healing chamber may vary with the implantation position and the health condition of the patient. Thus, customization, computer-aided design, and additive manufacturing have become important in the dental and healthcare industries. Many new configurations of implants have been proposed in the literature, such as porous [4], gyroid [5], and eagle-beak [6] shapes. It was shown, either experimentally or numerically, that these highly complex shapes performed better than implants manufactured by traditional machining techniques. However, designs are typically based on trial-and-error, and the relationship between the geometry of an implant and osseointegration is still subtle.

The history of bone ingrowth and tissue differentiation after insertion surgery is difficult to be observed experimentally. Thus, numerical methods are used to analyze the healing pattern in the initial stage of fracture healing [7,8]. The mechano-regulatory theory [9,10,11,12] is a well-accepted approach to predicting the osseointegration and tissue differentiation around implants [13]. Based on clinical experiments, Pauwels [14] first proposed that the formation of fibrous tissue and cartilage is characterized by the biophysical stimuli due to the distribution of bone elongation and hydrostatic pressure. Prendergast et al. [10] adopted deviatoric strain and fluid velocity as the biophysical stimuli. Lacroix and Prendergast [15] further improved the model, using poroelasticity material properties to simulate the mechanical behavior of bone tissue. The detailed healing pattern in the fractured bones were predicted accurately. Chou et al. [16] adopted an axisymmetric model to predict the length of the bone-implant contact and the bone volume formed in the healing callus around a set of implants using different micro-motions, healing callus size, and implant thread design. The results suggest that implant thread design is the most important factor of tissue differentiation.

In order to investigate the effect of implant geometry on the osseointegration and healing process, eight dental implants with distinct healing chamber designs were simulated by using the mechano-regulatory model in this work. Both the distribution of bones/tissues around the implants, and the solid/fluid stimuli will be analyzed and discussed. Moreover, the predicted bone ingrowth and performance of the implants will also be compared in the following sections.

## 2. Results

In order to validate the current model, we firstly reproduced the result of bone healing around a given implant shown in Chou et al. [16], which is a well-accepted research work using mechano-regulation algorithm. Here, all the settings used were following their works. The healing pattern on 28 day showed high similarity to Chou’s result as shown in Appendix A. Then, our models were further validated by comparing the history of bone ingrowth and cell differentiation around implants A and B (i.e., No. 1 and 3 implants used by Marin et al. [2]) with those in the animal test. Figure 1a shows the results of implant A. On the 4th day, the bones maturated from both sides, resulting in a thin band of the immature bones connecting all the threads of the implant. This feature shows good agreement with results found in the literature [17]. The resorption firstly appeared on the top surface of the callus region around the neck of the implant. This corresponds to the common feature of crestal bone loss in dental implants. Note that, although the mature and immature bones were present on the 4th day, Young’s modulus of these regions was still close to the granulation tissue, based on Equation (Equation 3). This was due to the low concentration of the mesenchymal stem cells at the early stage of cell migration. On the 21st day, immature bones were only found in the regions under the implant and the crest of the top threads. Resorption became significant with time, especially in the lower six healing chambers. There was no significant change in the distribution of the cell phenotypes after the 21st day. Young’s modulus of the mature and immature bones reached their maximum values on the 35th day, as the cell migration was completed.

Implant B had a similar bone ingrowth pattern to implant A, as shown in Figure 1b. However, mild resorption was found around the root of the healing chambers in the upper eight threads on the 21st day. Bone resorption became severe and extended to almost every implant thread by the 35th day.

As mentioned above, implants A and B corresponded to No. 1 and 3 implants used in the animal tests done by Marin et al. [2]. The wound-healing pattern they observed around implant A in beagle dogs, from the 21st and 35th day, are shown in Figure 2a,b, respectively. The predicted results generated by the current model are shown in Figure 2c,d, which are the enlarged images of the focused regions marked within the red rectangle in Figure 1a. Some instances of healing were also revealed in the simulations. For example, the resorption length was approximately half of the chamber length [2]; there were resorption bumps located at the middle of the lower flank of the healing chambers; and resorption typically formed at the upper-left corner of the healing chambers, as shown in Figure 2c,d. Details about the formation of resorption bumps will be examined later in the Discussion. In the case of implant B, mild bone resorption on the 21st day could be both observed experimentally and numerically, as shown in Figure 2e,g. The feature of resorption bumps was also revealed. On the 35th day, more bone resorption was discovered, and the resorption length was also half of the chamber length, as shown in Figure 2f,h. It is remarkable that the experimental and numerical results had very good agreement. Thus, the accuracy of the current model is verified.

Figure 1c shows the predicted healing pattern around implant C. On the 4th day, the granulation tissues which attached to the implant body differentiated into soft tissues. These soft tissues were surrounded by immature bones, and the mature bones filled in the rest of the callus region. Similar to the other implants, the resorption was initiated around the neck of implant C, on the 4th day. Although there was relatively less mature bone around the implant, bone resorption was absent in the healing chambers of implant C. It is of interest to note that the distribution of the phenotypes of cells remained consistent throughout the entire healing progress in this case, and only the material properties altered with the stem cell concentration due to the cell migration.

Implant D is an addictive-manufactured dental implant with eagle-beak shaped healing chambers designed by Lee et al. [6]. Since the design was modified from implant C [6], the healing patterns in the early stages were similar (see the patterns for the 4th day in Figure 1c,d). However, the non-uniform roots of implant D suppressed the formation of soft tissues. Similar to implant A, the mature bones grew from both sides, i.e., the cells origin and the healing chamber roots, during the 4th to 21st day. There were almost no immature bones or soft tissues left at the end of the healing process. In addition, the resorption regions in all healing chambers were so small, with a dimension of only two to three elements.

Now consider the performance of implants A through D by examining their bone-implant contact (BIC) and bone area (BA) on the 35th day, as shown in Table 1. The other four implants E to H will be discussed in the following section. Implant D gave the highest BIC and BA values among the four implants, 74% and 95%, respectively. The BIC value of implant C (64%) was higher than that of implants A (58%) and B (60%). However, the situation reversed when looking at BA values of implants A, B (74%, 71%), and C (68%). It is worth noting that the BIC and BA levels of implant C were mainly contributed to immature bones, while those of implants A, B and D were contributed to mature bones. Thus the osteointegration results of implant C might be overestimated, since the value of Young’s modulus of immature bone is six times lower than the value of mature bone. The results showed that implant D had the best osseointegration, and that healing chamber design can significantly affect the osseointegration and performance of dental implants.

## 3. Discussion

We next examine how the biophysical stimuli affect the healing pattern and tissue differentiation around implants. Figure 3 and Figure 4 show the contour plots of solid and fluid stimuli (the first and second terms in Equation (Equation 1), respectively) for implants A through H on the 4th, 21st, and 35th day. In general, high solid and fluid stimuli were observed in the regions under the tip of implant due to the vertical loading. In most cases, both stimuli decreased with time, and thus, bone resorption (S≤0.010) primarily occurred later on.

As for implant A, the healing was dominated by solid stimulus, since the pattern of soft tissues, the thin band of immature bone on the 4th day, and the resorption regions on the 21st and 35th day (see Figure 1a) were in accordance with that of the solid stimulus, as shown in Figure 3a. By contrast, the healing around implant B was dominated by fluid stimulus. It can be observed in Figure 4b that the pattern of fluid stimulus matched the immature bone regions on the 4th day, and the resorption regions on the 21st and 35th day (see Figure 1b).

Both implants A and B led to severe bone resorption. This was caused by the healing chambers’ poor design and relatively greater depth. From the perspective of solid stimulus, the thread crests provided the major fixation of the implant against the applied vertical load. Thus, the regions near the root of the healing chambers were less strained, leading to a low solid stimulus (see the bumps in dark blue on the 21st day in Figure 3a,b). The shape of the over-sized healing chambers resulted in undesired resorption and poor osseointegration.

It is significant that implant C lead to the most stable healing pattern among all the cases. Implant C had the smallest healing chambers in the current study. This allowed the fluid to flow into the entire chambers, resulting in high fluid stimulus ranging from 0.27 to 3.0, as shown in Figure 4c. In addition, high solid stimulus (greater than 1.0) can be found in Figure 3c due to the stress concentrated at the small threads. Both high stimuli lead to immature bones and soft tissues covering the implant. The feature of soft tissues covering the bone-implant interface was also found in the literature [18]. Moreover, the large amount of immature bones occurred around the implant, reducing the strength of the entire implantation system, as the Young’s modulus of immature bones was only one sixth the value of the mature bones.

The depth of the healing chambers of implant D was relatively greater than that of implant C, but shallower than that of implant A. The values and contours of both stimuli of implant D were similar to those of implant C on the 4th day, as shown in Figure 3c,d, and Figure 4c,d. However, the healing pattern of implant D became similar to that of implant A on the 21st day, e.g., the cartilage and fibrous tissues were absent. On the 35th day, both stimuli of implant D, in most of the healing chambers, decreased to suitable values ranging from 0.01 to 0.27. Therefore, no severe bone resorption occurred, resulting in the formation of mature bones. This healing chamber design [6], with its adequate depth and eagle-beak shape, significantly improves the resulting BA and BIC, and provides for better osseointegration of the implant.

In order to understand the influence of the design of healing chambers on solid, fluid stimuli, and osseointegration, we will now discuss the performance of implants E to H, which were modified from implants A to D. In this section, the implants will be divided into three groups to reveal the influence of the slope of flank (Group1: A, E, F), the length of root and crest (Group2: B, G), and the depth of healing chamber (Group3: C, D, H) on the performance of implants, as shown in Figure 1.

Implants E and F were based on implant A, and yet, showed increasing slopes of the upper flanks of each healing chamber (i.e., $\phi_{A}>\phi_{E}>\phi_{F}$). The slopes on the upper and lower flanks of implant E were the same, while implant F had steeper upper flanks than its lower flanks. The depth of healing chambers in implant G were similar to that of implant B, however, the roots (*r*) and crests (*c*) of implant G were flat (i.e., $r_{B}<r_{G}$, $c_{B}<c_{G}$). Finally, implant H had a greater depth in the upper half of the healing chambers compared with implant C (i.e., $d_{C}<d_{H}$). This also had inverse eagle-beak shaped healing chambers (upside down to the design of implant D). The tissue differentiation of implants E to H on the 4th, 21st, and 35th day are shown in Figure 1e–h. The corresponding solid and fluid stimuli of implants E to H are also shown in Figure 3 and Figure 4e–h, respectively.

Now consider implants A, E, F. It was found that in the 35th day, steeper slopes in the upper flanks (i.e., $\phi_{F}<\phi_{E}<\phi_{A}$) generally resulted in higher solid and fluid stimuli, especially in the region under implants (please see Figure 3a,e,f and Figure 4a,e,f. This was because the smaller the angle of ϕ, the less fixation and resistance is provided by the surrounding bones (ϕ is the angle between the orientations of the upper flanks and the direction of the applied force) Thus, there was less bone resorption in the lower healing chambers, and more immature bones in the region under implant F on the 35th day (as shown in Figure 1a,e,f).

Table 1 shows the BA and BIC values of implants A, E, and F. Note that, the calculation of BA/BIC focuses on the region we concerned about (ROI), which mainly contributes to the performance of osseointegration near threads. It can be observed that implants with steeper slopes in the upper flanks (ϕ) provide greater values of BA and BIC in 35th day, leading to better osseointegration, e.g., implant F had the highest values (84% and 71%, respectively) among the three implants, while implant A had the lowest values (74% and 58%, respectively). These findings are in good agreement with the reported literature [16]. Next, the model can also predict the overall strength of the surrounding bones, the averaged Young’s modulus evolution process for the eight implants were shown in Figure 5a. Although the values of BA/BIC of implant F were the highest in Group1, the overall averaged Young’s modulus in callus region showed with an opposite trend. As mentioned above, this is caused by the relatively higher biophysical stimulus, leading to the exist of fibrous tissue, cartilage and immature bone with lower Young’s modulus in the region under implant F.

By comparing the shape of healing chambers in implants B and G, the effect of the roots and crests can be revealed. It can be observed that implant G maintains a greater value (>0.27) of the fluid stimulus in most of the regions until the 21st day, as shown in Figure 4g. Due to the high biophysical stimulus, there was less bone resorption in implant G on the 21st and 35th day (Figure 1g), giving great values of BA and BIC (86% and 75%, respectively), compared to that of implant B. Thus, in the cases of implants with similar depths, the flat roots and crests of the healing chambers led to better osseointegration. Furthermore, according to Figure 5a, bone tissue maturated faster in the case of implant B (orange line) than it of implant G (black line), leading to to overall higher Young’s modulus in the surrounding bones.

Now consider implant H, which has a greater depth in the upper half of the healing chambers, compared with implant C. It is of interest to note that, such differences do not seem to alter the contours of the healing pattern, the solid stimuli, or the fluid stimuli of implant H, compared to those of implant C, as shown in Figure 1, Figure 3 and Figure 4c,h. By contrast, the shape of implant D (inverse of implant H) has a greater depth in the lower half of the healing chambers compared with implant C, and the history of the healing pattern was dramatically changed, as is mentioned in the Results. Thus, it can be concluded that in the cases of the considered implants which were subjected to vertical loading, the influence of the depth of the lower half of the healing chambers is relatively more significant than the depth of the upper half of the healing chambers.

Although the healing patterns of implants C and H were similar, the BA and BIC numerical data of implant H were improved, as shown in Table 1. This indicates that the depth (*d*) of the healing chamber is crucial for the performance of implants. Furthermore, when comparing the cases of implant D with H, it can be seen that implant D resulted in a greater value of BA, and had more mature bone implant contact. Next, by examine the averaged Young’s modulus evolution in Group3 (see Figure 5a), it was observed that the averaged Young’s modulus of implant D was also the highest among the eight implants. Moreover, Figure 5b shows the averaged Young’s modulus distribution of each elements in callus region for implants C, D and H. The results showed that values of the most of the elements in implant C and H were in the range of 0 to 2 MPa, while those in implant D were in the range of 4 to 6 MPa. This shows that although implant H has high BA/BIC values, the overall strength of the surrounding bone of implant H was still relatively lower than it of implant D. This confirms, once again, that implants with steeper upper flanks lead to better osseointegration, as mentioned above.

It has been proved by in vivo and in vitro animal test that thread design has a great influence on the osseointegration [19,20,21]. Scarano et al. [22,23] found that vessel formed in the concavities of the threads (i.e., inside the healing chambers) rather than the convexities, and this was positively correlated with bone formation rate. Bone healing of this type was also reproduced in the current study, where implants A, E, and G resulted in the initiation of mature bones from the root of the healing chambers in the early stage (Day 4) of bone healing, as shown in Figure 1a,e,g. Moreover, several simulation studies have focused on the biomechanical stimulus of bones, in order to eliminating stress concentration and reducing bone resorption around implants [24,25,26,27]. However, uniform material properties of bones around the implants were typically assumed; cell differentiation and fluid stimulus was also ignored. Thus, the current study resolved these problems and gave a direct analysis for osseointegration.

The current study reproduced the bone healing features of the work done by Marin et al. [2], and provided possible design guideline for the geometry of the healing chamber of implants. However, only vertical applied force can be considered since a simplified 2D axisymmetric model was used in the current work. This simplification was necessary as mechano-regulation algorithm was typically computationally demanding, and many design parameters in healing chambers should be processed. This issue can be resolved by incorporating 3D FEM model and regarded as the future work.

## 4. Materials and Methods

The current model was built in two-dimensional axisymmetry using the commercial finite element (FE) package, ANSYS. It consisted of a Ti-6Al-4V implant, a cortical layer, cancellous bones, and callus. The geometrical detail is shown in Figure 6a. The implant was meshed by 2D four-node solid structure elements, with the ANSYS built-in element type, PLANE42. The remaining parts of the geometry, e.g., the calluses, cortical bones, and cancellous bones, were meshed by 2D four-node coupled pore-pressure mechanical solid element, with the element type CPT212. Elements of this type enable the calculation of the fluid velocity and pressure in the pores of the bones while under external mechanical loads. A mesh size of 0.05 mm was set to allow for an accurate and stable solution, while the detail of the complex configuration could also be captured.

An average displacement of 8μm along the −y axis was applied at nodes on the top surface of the implant. This value corresponded to the displacement resulting from the biting force (≈100 N) of beagle dogs [28], and thus, our simulation results could be validated by the data obtained from the animal tests done by Marin et al. [2] (the trial followed the approval of the bioethics committee for animal experimentation at the Universidade Federal de Santa Catarina, Brazil) The displacement degree of freedom along the *y* axis at the bottom nodes and along the *x* axis in the bottom-left node were constrained, as shown in Figure 6a. The axisymmetric boundary condition was set at the nodes lying on the *y* axis, marked as a dot-dashed line. The callus region located around the implant was where the bone/tissue differentiation occurred. The boundary, marked with a dashed line, shows the cells’ origin, and is defined as the source of diffusion of the mesenchymal stem cells, i.e., the stem cell concentration was set as 100%. The interface between the implant and callus was set to allow sliding with a friction coefficient of 0.3 [29]. The material properties used in the current study are shown in Table 2 [16].

There were a total of 35 iterations in the simulation, corresponding to the healing period in the animal tests [2]. In the current study, each iteration represented one healing day. The iterative procedure was based on the procedure of Chou et al. [16]. A MATLAB user-defined function was developed to automate the modification of the FE model.

The solid and fluid stimuli considered in the mechano-regulation model corresponds to two important values [10]: the octahedral shear strain γ and the relative fluid/solid velocity $\nu =\sqrt{\nu_{x}^{2}-\nu_{y}^{2}}$, where $\nu_{x}$ and $\nu_{y}$ are the fluid velocity in the transverse and normal directions. The phenotype of the cells of each element in the callus region for the next step, i.e., (i+1)th iteration, was determined by the cell stimulus factor *S*, which is related to γ and ν, such that
(1)$$S=\frac{\gamma }{a}+\frac{\nu }{b}$$
where solid stimulus constant a=0.0375 and fluid stimulus constant b=3μm/s [36]. The ranges of the value of *S* and the corresponding phenotypes of cell, including fibrous tissue (S>3), cartilage (3≥S>1), immature bone (1≥S>0.266), mature bone (0.266≥S>0.010), and resorption (0.010≥S), can be specified. Each phenotype has a set of material properties, such as Young’s modulus, Poisson’s ratio and permeability, as shown in Table 2 [16,30]. Thus, the material settings in each element in the callus region were required to be updated for the next iteration.

Two issues needed to be considered when updating the material settings for the next step, the (i+1)th iteration. The first issue was the mesenchymal stem cell migration [11], which is related to the level of transition from the granulation tissue to cells of different phenotypes. It was modeled as a random anisotropic diffusion [15]. The concentration of stem cells n increased in each successive step of iterations, governed by
(2)$$\frac{dn}{dt}=D\nabla^{2}n$$
where *t* is time; *D* is the diffusion coefficient with the unit of square meter per day. Here, the diffusion coefficient *D* was determined so that the entire healing callus reached the maximal cell concentration after 35 day [2,12]. The effective material property $X_{mix}$ in the next iteration, such as Young’s modulus, Poisson’s ratio and permeability, depended on the concentration of the stem cells $n_{i}$ in the *i*th iteration, that is
(3)$$X_{mix}=\frac{n_{max}-n_{i}}{n_{max}}X_{g}+\frac{n_{i}}{n_{max}}X_{d}$$
where $n_{max}$ is the maximum concentration of cells; $X_{g}$ and $X_{d}$ are the material property of the granulation tissue and the differentiated tissue, respectively. With this linear combination, the progress of tissue transition could be modeled.

The second issue when updating the material settings for the next iteration was the application of the smoothing procedure. The material property $X_{mix}$ obtained in Equation (Equation 3) needed to be averaged with those in the previous nine iterations in order to avoid numerical instability and sudden changes. The material properties for the next iteration could be written as
(4)$$X_{i}=\frac{1}{N}(X_{mix}+X_{i-1}+X_{i-2}+\ldots +X_{i-(N-1)})$$
where *i* is the number of iteration. For the iteration of i<9, the average was made from the *i*th iteration to the first.

In the current study, four implants labeled A to D were considered first. Implants A and B corresponded to No. 1 and 3 implants used in the animal test done by Marin et al. [2]. Implant C was similar to the commercial ITI (Institute Straumann AG, Waldenburg, Switzerland) solid cylindrical screwed implant, number 033.512S. The healing chambers of implant D were eagle-beak shaped, which were modified from implant C in the author’s previous work [6]. Such complex configuration was expected to promote osseointegration. Next, implants E to H, which were modified from implants A to D, were also discussed to reveal the influence of healing chamber geometry on osseointegration.

Two parameters were defined for the histomorphometric evaluation of the performance of implants: bone-implant contact (BIC), and bone area (BA). The region of interest (ROI) of BIC is the total length of upper/lower flank and the root of all the threads in the implant. ROI of BIC in one of the threads are illustrated with red solid lines in Figure 6b. BIC is defined as the percentage of the total length of the interface between the implant elements and the mature/immature bone elements in the ROI. Similarly, ROI for BA is the total area between all the threads (illustrated as shaded area). The definition of BA is the percentage of the total area of the mature/immature bone elements in the ROI. It is expected that the higher the BIC and BA values, the better the osseointegration of an implant [16].

## 5. Conclusions

In the current study, a 2D axisymmetric finite element model with a mechano-regulatory algorithm was developed to study the bone ingrowth and tissue differentiation around eight dental implants, A to H, each with different healing chamber designs. The predicted healing patterns in implants A and B were verified by the data obtained in the animal test done by Marin et al. [2], showing very good agreement. The performance of the implants was then evaluated by bone area (BA) and bone-implant contact (BIC). The results showed that severe bone resorption occurred due to the over-sized and sharp configuration of the healing chambers. The commercial ITI implant (implant C) with small healing chambers led to high solid and fluid stimuli retention, immature bones and soft tissue covering the implant, moreover, resulting in the most steady healing pattern with no significant change on the phenotype of cells. Implant D, which was suitable for additive manufacturing, had eagle-beak shaped healing chambers, and showed remarkable osseointegration, based its BA and BIC values.

Additional implants were discussed to reveal the influence of the designs of the healing chamber on both the stimuli and the healing patterns. Implants A, E, and F showed that steeper upper flanks result in less resorption and better osseointegration. The results of implants B and G revealed that flat roots of healing chambers lead to better osseointegration. The cases of implants C, D and H show that the depth of the upper half of the healing chambers has a significant effect on the healing pattern. The healing chamber design of implant D matched all the design criteria mentioned above, such as steep upper flanks, relatively flat roots, and adequate depth. Thus, implant D led to the best osseointegration among all the implants considered in the current work.

The mechano-regulatory tissue differentiation model developed in the current work is rapid, accurate, and customizable for any given dental implant with an axisymmetric cross-section. The results generated here can also serve as design guidelines and our model is able to provide “possible trends” for improving the performance of dental implants under the specific boundary conditions considered in the current work.

## Figures and Tables

**Figure 1 ijms-21-09205-f001:**
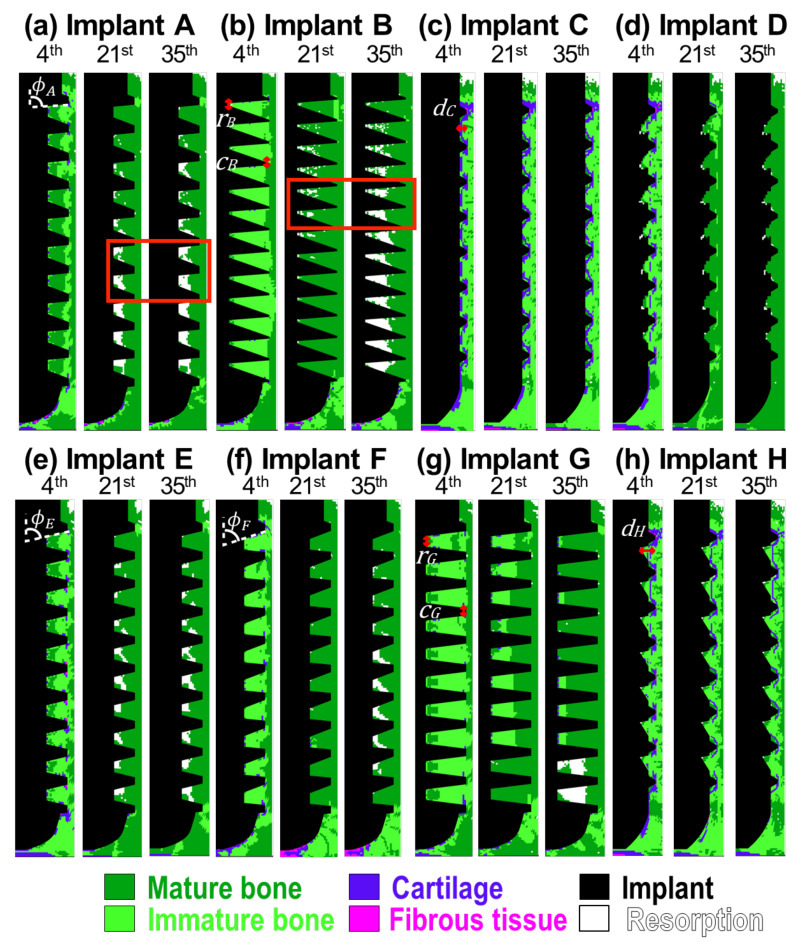
(**a**–**h**) The predicted history of the bone ingrowth and cell differentiation around implant A to H.

**Figure 2 ijms-21-09205-f002:**
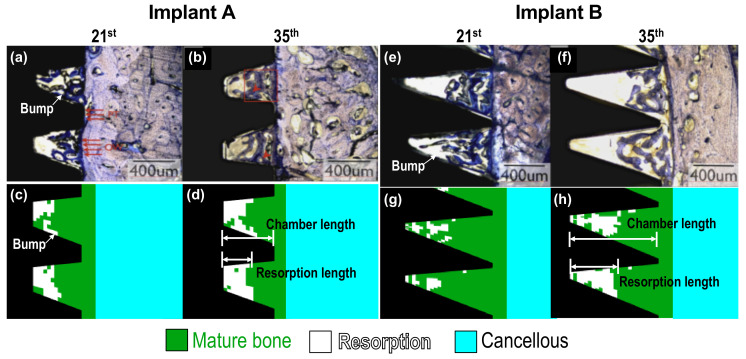
The wound-healing pattern around (**a**,**b**) implants A and (**e**,**f**) implant B on the 21st and 35th day observed in the animal experimental done by Marin et al. [2]. Zoom-in on the region marked in red in Figure 1, showing the predicted patterns of (**c**,**d**) implant A and (**g**,**h**) implant B.

**Figure 3 ijms-21-09205-f003:**
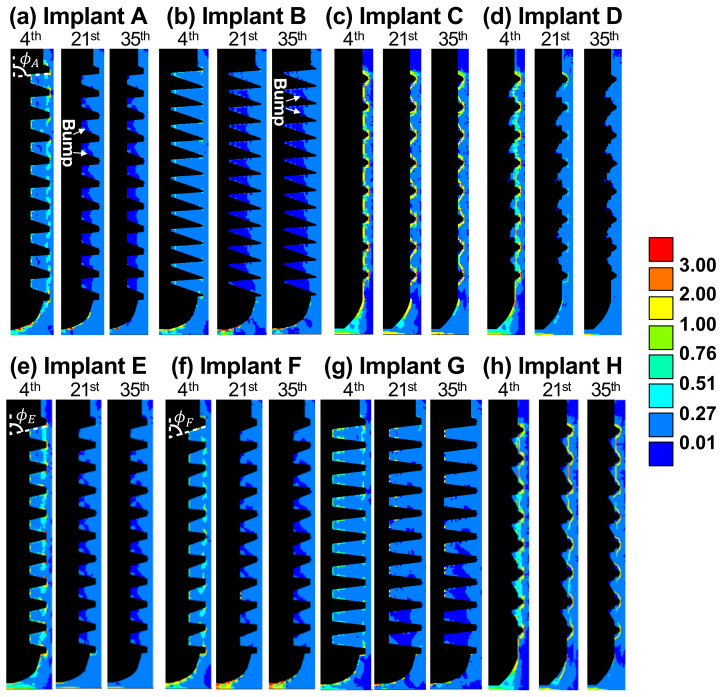
(**a**–**h**) The contour plots of solid stimulus for implants A to H, on the 4th, 21st, and 35th day.

**Figure 4 ijms-21-09205-f004:**
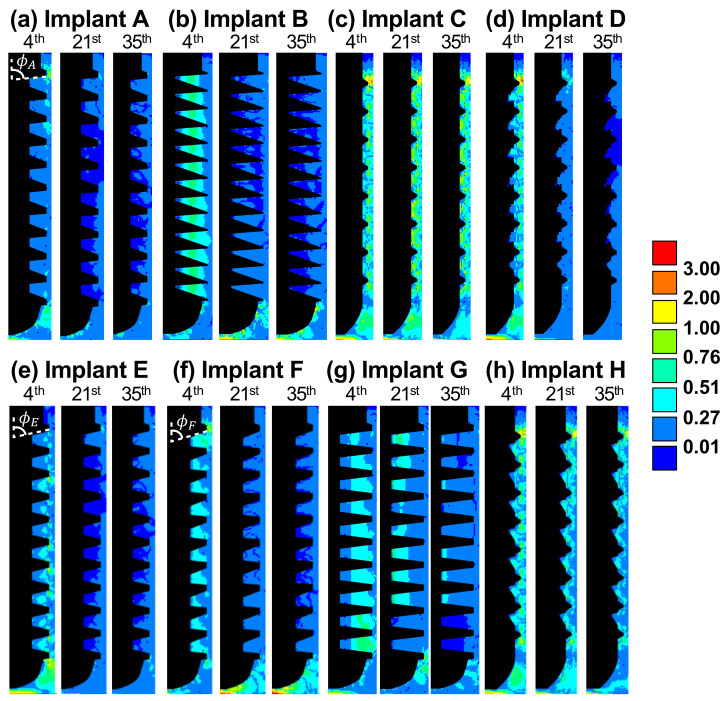
(**a**–**h**) The contour plots of fluid stimulus for implants A to H, on the 4th, 21st, and 35th day.

**Figure 5 ijms-21-09205-f005:**
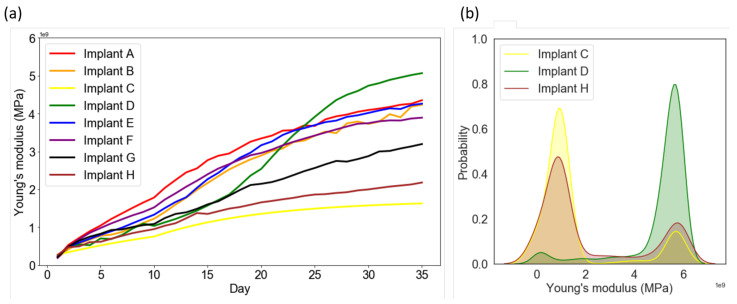
(**a**) The overall averaged Young’s modulus evolution in the callus region for implant A–H. (**b**) The Young’s modulus distribution of each element in the callus region for implants C, D, H.

**Figure 6 ijms-21-09205-f006:**
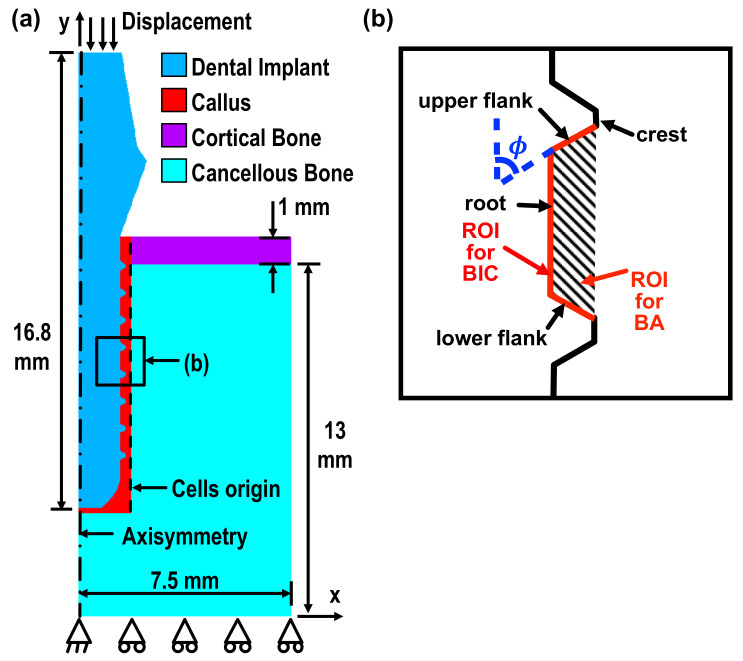
The schematic plots of (**a**) the model with boundary conditions and (**b**) ROI of BIC and BA in one of the threads.

**Table 1 ijms-21-09205-t001:** The values of BA and BIC of implants A to H.

Implant	A	B	C	D	E	F	G	H
Bone Area (BA%)	74	71	68	95	80	84	86	85
Bone implant contact (BIC%)	58	60	64	74	69	71	75	85

**Table 2 ijms-21-09205-t002:** The material properties used in the current work [11,30,31,32,33,34,35].

Tissue Phenotype	Young’s Modulus	Poisson’s Ratio	Permeability
	(MPa)		(m${}^{4}$/Ns)
Granulation tissue	1	0.17	$1\times 10^{-14}$
Fibrous tissue	2	0.17	$1\times 10^{-14}$
Cartilage	10	0.17	$5\times 10^{-15}$
Immature bone	1000	0.30	$1\times 10^{-13}$
Mature bone	6000	0.30	$3.7\times 10^{-13}$
Cortical bone	20,000	0.30	$1\times 10^{-17}$
Cancellous bone	6000	0.30	$3.7\times 10^{-13}$
Ti-6Al-4V	113,000	0.30	N/A

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
