# Peer review of "Healing Pattern Analysis for Dental Implants Using the Mechano-Regulatory Tissue Differentiation Model"

_ijms, 2020, doi:10.3390/ijms21239205_

Round 1
Reviewer 1 Report
Comments
In the present work, the authors showed several healing patterns for dental implants based on the mechano-regulatory model and try to show the improvement of the bone ingrowth and osseointegration using the depth, slope of upper flanks, and the shape of roots. However, more detailed data are necessary to discuss these.
(1) It seems that the distributions of mature and immature bone are important.
The difference between these distributions for the eight cases (especially C, D, G, and H) and their time dependence should be discussed in addition to BA and BIC.
(2) More detailed discussions on Φ and d are needed using the Φ (and d) dependence of BA and BIC, if possible.
(3) On page 7, line 156: the reason for the overestimation (?) must be clearly described because authors discuss the healing pattern, BA and BIC using the result of the implant C, for example, on page 8, line 178 and page 10, line 236.
Author Response
Response_ijms-980391
REVIEWER 2
In the present work, the authors showed several healing patterns for dental implants based on the mechano-regulatory model and try to show the improvement of the bone ingrowth and osseointegration using the depth, slope of upper flanks, and the shape of roots. However, more detailed data are necessary to discuss these.
- It seems that the distributions of mature and immature bone are important. The difference between these distributions for the eight cases (especially C, D, G, and H) and their time dependence should be discussed in addition to BA and BIC.
A: Thanks for the reviewer’s suggestions. We have added fig.6(a) in the manuscript to illustrate the time dependence of changes in overall averaged Young’s modulus among the eight implants due to the varying tissue phenotype, solid and fluid stimuli in this manuscript. For example, “bone tissue maturated faster in the case of implant B (orange line) than it of implant G (black line), it had already dominated the healing chamber on the 21st day, which leads to higher Young’s modulus in the surrounding bones.” The texts were added in Line 225-236, Line 239-245, and Line 259-265 to address this issue.
- More detailed discussions on Φ and d are needed using the Φ (and d) dependence of BA and BIC, if possible.
A: Thanks for the reviewer’s suggestions. We have discussed the dependence of BA and BIC of Φ and d in Line 225-236 (colored in red). In Group1 (implant A, E, F), the results reveal that the steeper slopes in the upper flanks (bigger Φ), the lower value of BA/BIC; Then, by comparing among Group3 (implant C, D, H), it was found that the depth (d) of the healing chamber is crucial for the performance of implants. The performance of BA/BIC value in implant C can be improved by increasing the depth (d) in the healing chamber. In addition, the relationship between the averaged Young’s modulus and Φ, d was also be discussed in this article (in Line 259-265, colored in red)
- On page 7, line 156: the reason for the overestimation (?) must be clearly described because authors discuss the healing pattern, BA and BIC using the result of the implant C, for example, on page 8, line 178 and page 10, line 236.
A: Thanks for the reviewer’s suggestions. We have explained the reason for the overestimation more clearly in Line 105-111. The reason is that Young’s modulus of immature bone is 6 times less than mature bone in our model, and thus, the strength of implant C (which is dominated by immature bone) processes a lower strength compared to implants A and D. To illustrate this issue, we have added fig.6(b) in manuscript, showing Young's modulus distribution of each element in callus region for implants C, D, H. The results showed that the values of most elements in implant C and H were in the range of 0 to 2 MPa, while those in implant D were in the range of 4 to 6 MPa. This shows that although implant H has high BA/BIC values, the overall strength of the surrounding bone of implant H was still relatively lower than it of implant D (in Line 259-265).

Reviewer 2 Report
General comments
Although the use the mechano-regulatory tissue differentiation based on biophysical stimuli and healing patterns is of great scientific interest, the 2D FEM approach and a vertical loading is too simplistic since do not accounting for the complex deformation state that is generated in peri-implantar bone during the in-vivo masticatory function (due to maxillary and/or mandibular bone orthotropy as it can be found in literature and in some already published papers). The proposed interesting simulation only considering 100 N vertical loading without a specific in vivo validation experimental part remains a scholastic exercise (the strains distribution on the mandibular or maxillary bone where implants are positioned in the cited experimental by Marin et al. is not known and not reproduced by the proposed model). A more correct approach (as it has been well described in existing literature on FEM modeling of dental implants and peri-implantar strain distribution) should first be validated for strain distribution under masticatory loadings in a FEM model of the maxillary or mandibular healthy bone. Once validated, the model with the different surface geometries implants can be simulated (eventually considering a sub-modeling of the single areas of interest).
The results of the simulations In the present form can not be compared to existing in vivo experiments. Due to the simplistic 2D approach, the proposed mechano-regulatory model can be applied only to give generic (and not so specific) indications on "possible" trends of bone tissue differentiation in the 8 different implants considered in the study.
Author Response
Response_ijms-980391
REVIEWER 1
1. Although the use the mechano-regulatory tissue differentiation based on biophysical stimuli and healing patterns is of great scientific interest, the 2D FEM approach and a vertical loading is too simplistic since do not accounting for the complex deformation state that is generated in peri-implantar bone during the in-vivo masticatory function (due to maxillary and/or mandibular bone orthotropy as it can be found in literature and in some already published papers). The proposed interesting simulation only considering 100 N vertical loading without a specific in vivo validation experimental part remains a scholastic exercise (the strains distribution on the mandibular or maxillary bone where implants are positioned in the cited experimental by Marin et al. is not known and not reproduced by the proposed model). A more correct approach (as it has been well described in existing literature on FEM modeling of dental implants and peri-implantar strain distribution) should first be validated for strain distribution under masticatory loadings in a FEM model of the maxillary or mandibular healthy bone. Once validated, the model with the different surface geometries implants can be simulated (eventually considering a sub-modeling of the single areas of interest).
A: Thank you for the reviewer’s suggestion. The reviewer pointed out many important issues in the current model settings, such as the loading conditions and verification methods of the model. The corresponding answers and responses are as follows:
Loading conditions: The current work focuses on the relationship between design parameters in healing chambers and performance. Mechano-regulation algorithm requires a lot of computing resources, and thus, bone healing around the dental implants was simulated by a simplified 2D axisymmetric model. This makes that the lateral force cannot be simulated in the current model. This limitation is addressed in Line 267-272, and this issue can be resolved by incorporating the 3D FEM model, which is regarded as future work in this manuscript.
Verification of the model: The reviewer suggested to validate the model by comparing the strain distribution in the existing literature on FEM modeling. However, the literatures where strain distribution was considered are typically correlating the material properties with bone density rather than cell phenotypes. Thus, here we reproduced the cell-phenotype distribution of a dental implant generated in Chou et al. 2013 [1], which is also a research work using mechano-regulation algorithm published in a well-accepted journal, Journal of Biomechanics. Our result showed high similarity to Chou’s results as shown in the figure below. It is worth noting that, the tissue phenotypes in both models were labeled in different colors. We have added our reproduced result and the corresponded descriptions in Line 105-111 in the manuscript, and a detailed explanation was written in supplementary.
2. The results of the simulations in the present form can not be compared to existing in vivo experiments. Due to the simplistic 2D approach, the proposed mechano-regulatory model can be applied only to give generic (and not so specific) indications on "possible" trends of bone tissue differentiation in the 8 different implants considered in the study.
A: Thanks for the reviewer’s suggestion. We have rephrased our texts to state that our model is able to provide “possible trends” for improving the performance of dental implants under the specific boundary conditions considered in the current work. We have revised the corresponding description in the article (Line 295-297).

Round 2
Reviewer 1 Report
Comments
The revised manuscript has been modified according to the comments. I recommend the publication of the manuscript (980391) to IJMS.
Author Response
Dear Reviewer,
Thank you very much!
Reviewer 2 Report
The Authors correctly reply to criticisms by adding the justifications in the text
Author Response
Dear Reviewer,
Thank you very much!